# Hepatitis B vaccine uptake and associated factors among pregnant women attending antenatal care at Gulu Regional Referral Hospital, Northern Uganda

**Linda Mercy Akello**[1]*, **Jimmy Osuret**[2‡], **Jovan Galiwango**[2‡], **Amos Deogratius Mwaka**[1,3], **Christopher Garimoi Orach**[1,4]

1 Department of Public Health, Faculty of Medicine, Gulu University, Gulu, Uganda, 2 Department of Disease Control and Environmental Health, School of Public Health, Makerere University, Kampala, Uganda, 3 Department of Internal Medicine, Faculty of Medicine, Gulu University, Gulu, Uganda, 4 Department of Community Health and Behavioral Sciences, School of Public Health, Makerere University, Kampala, Uganda

☯ These authors contributed equally.
‡ These authors also contributed equally.
* mercylindaakello@gmail.com

## Abstract

### Background

Hepatitis B virus (HBV) infection is a global public health concern. Uganda has a high prevalence of HBV infection among pregnant mothers. This study aimed to determine HBV vaccine uptake and associated factors among pregnant women attending Antenatal Care (ANC) at Gulu Regional Referral Hospital (GRRH).

### Methods

This was a cross-sectional study. Data were collected from 430 participants who were selected by systematic sampling. Data were collected using an electronic questionnaire built on the Kobo Collect server and analyzed using STATA version 16.0. Prevalence ratios (PR) were calculated using a modified Poisson regression analysis to determine the association between HBV vaccine uptake and the predictors.

### Results

More than half (53%:229/430) of the pregnant mothers had received at least a dose of the HBV vaccine. Factors statistically significantly associated with HBV vaccine uptake included age (APR = 1.10, 95% CI: 1.02–1.18), knowledge about HBV (APR = 1.06, 95% CI: 1.01–1.10), and healthcare worker communication (APR = 1.17, 95% CI: 1.04–1.32).

**Data availability statement:** The minimal dataset is available as Supporting Information files.

**Funding:** The author(s) received no specific funding for this work.

**Competing interests:** The authors have declared that no competing interests exist.

**Abbreviations:** ANC: Antenatal Care; CDC: Center for Disease Control; GRRH: Gulu Regional Referral Hospital; HBeAg: HBVe-Antigen;HBV: Hepatitis B; HBV: HBV virus; HBV-BD: Hepatitis B-Birth Dose; IRBs: Institutional Review Boards; IDI: Infectious Disease Institute; MTCT: Mother-to-Child Transmission; PI: Principal Investigator; WHO: World Health Organization.

## Conclusion

More than half of pregnant mothers had received at least a dose of the HBV vaccination. Age, knowledge, and healthcare communication influenced vaccine uptake. There is a need to provide adequate health education regarding the importance of completing the HBV vaccine to pregnant women and community members to improve vaccine outcomes.

## Introduction

Globally, Hepatitis B (HBV) infection poses a significant health threat, particularly due to its increased risk of liver cancer and death [1]. In 2022, the World Health Organization (WHO) estimated that 254 million people were living with chronic HBV infection, with more than 1.2 million new cases reported annually. This led to over 1.1 million deaths in the same year, primarily due to cirrhosis and hepatocellular carcinoma [2]. About 65 million people in the WHO African region live with chronic HBV infection, with over 60 million of them residing in Sub-Saharan Africa (SSA) [2,3]. The majority of individuals with chronic HBV infections acquired the infection during the perinatal period [4]. Individuals who get infected during the perinatal period are 95% likely to become chronic carriers compared to those who acquire the infection during adulthood [5]. Pregnant mothers who test positive for both Hepatitis B surface antigen (HBsAg) and Hepatitis B e antigen (HBeAg) have a high risk (70–90%) of transmitting the infection to their newborns [6,7]. Therefore, pregnant women should be screened for HBsAg and Hepatitis B e-antigens during antenatal care, and mothers who are positive for both tests need to have their infants vaccinated for HBV at birth. [8,9]. In 2022, an estimated 1,250 Ugandans died of HBV-related complications, while approximately 6% of Uganda's population remained chronically infected [10]. According to the Uganda population-based HIV impact (UPHIA) assessment, the prevalence of HBV infection among adults aged 15–64 years was 4.1%, with the northern region having the highest prevalence rate of 4.6% [11]. A community-based study in Gulu City (formerly a Municipality) found that the prevalence of HBV in the general population was 17.6% [12]. A study among pregnant women in Gulu revealed a prevalence of HBV infection of 12% [13]. Data from the Gulu Cancer Registry (established in 2014) showed that the incidence and burden of liver cancer in the catchment area of the registry were 12.8 per 100,000, which was much higher than the liver cancer incidence and burden in the Kampala region of 10.0 per 100,000, where the incidence and burden of HBV infection are lower. The northern region has a higher prevalence of HBV (5.4%) compared to the central region, which is the catchment area of the Kampala Cancer Registry, with an HBV prevalence of approximately 2.3% [14,15].

HBV is a viral infection that can be transmitted through contact with infected body fluids like blood, saliva, vaginal fluids, and semen. It can also be passed from mother to baby during childbirth (transplacental transmission), via a breach in the placental barrier (intrauterine transmission), during delivery due to exposure to infected cervical secretions and blood (natal transmission), and postnatally through care or breast

milk, especially if proper immuno-prophylaxis is not initiated at birth or continued as recommended [16–18]. Widespread vaccination has proven to be the primary tool and most effective method of preventing HBV infection, reducing its prevalence, and preventing associated complications [19]. Furthermore, maternal vaccination plays a crucial role in protecting both mothers and their vulnerable infants [20]. The vaccine is highly effective, offering up to 95% protection against chronic infection if the dosage is completed [21]. According to the WHO, pregnant women should receive the vaccine at 0, 1, and 6 months to provide full protection for the mother and unborn child. If an alternative schedule is considered, a fourth booster dose is given at 1 year to provide maximum, long-term protection [22,23]. During antenatal care (ANC), pregnant mothers are advised to undergo screening and testing for diseases, including HBV. Following results from screening and testing, it is recommended that they receive vaccinations to provide prophylactic protection [24,25]. The WHO has proposed a global target for viral hepatitis control of a 30% reduction in new infections of Hepatitis B and C by 2020, and a 90% reduction by 2030. A mortality target of 10% reduction in deaths due to Hepatitis B and C by 2020, and a 65% reduction by 2030, has also been proposed [26]. These targets could be achieved through dramatic increases in the coverage of key preventive interventions including vaccination of the at-risk groups [27]. Vaccination is the most effective measure to reduce the global incidence of HBV infection in comparison to other healthcare interventions [28,29]. The Uganda Ministry of Health (MOH) launched a nationwide free HBV screening program in 2015, coupled with strong community mobilization, awareness campaigns, and vaccination efforts [30]. Furthermore, in Uganda, HBV vaccination is integrated into routine childhood immunization schedules, often starting at birth, as part of the broader efforts to reduce HBV transmission. In addition to infants, high-risk groups such as healthcare workers, people with multiple sexual partners, and individuals with compromised immune systems are encouraged to get vaccinated [31]. There are guidelines for HBV prevention, care, and treatment including vaccination, risk behavioral change, and universal precautions for infection prevention and control [32]. Preventing perinatal HBV transmission is an integral part of attaining Sustainable Development Goals (SDGs) three which aims at having good health and well-being by ending the epidemics of AIDS, tuberculosis, malaria, and neglected tropical diseases and combating hepatitis, water-borne diseases, and other infectious diseases by 2030, thereby reducing maternal and neonatal mortality [33]. Furthermore, the initiative supports SDG 5 (Gender Equality) by ensuring that women receive essential health services, empowering them to make informed decisions about their health and that of their children [34]. Despite these efforts, HBV testing and treatment are still inadequate among pregnant women in Uganda, particularly in northern Uganda [35,36]. A study in Gulu city has shown that of 397 pregnant women, 47 (11.8%) tested positive for hepatitis B surface antigen (HBsAg), with 7 (14.9%) of the HBsAg-positive mothers also being HBeAg positive. The study also showed that the highest HBsAg positivity rate was (20%) among younger women aged 20 years or less [13]. The objective of this study was to assess HBV vaccine uptake and associated factors among pregnant mothers attending ANC at a public hospital, the Gulu Regional Referral Hospital (GRRH) in northern Uganda, to generate evidence to guide the development of targeted interventions and policy to improve vaccine uptake and implement effective HBV prevention strategies.

## Methods

### Study design and setting

We conducted a cross-sectional study among pregnant women attending antenatal care at Gulu Regional Referral Hospital (GRRH). GRRH is a public hospital located in Gulu City, Northern Uganda, and serves as a referral facility for the districts of Agago, Amuru, Gulu, Kitgum, Lamwo, Nwoya, Omoro, and Pader [37]. Gulu Regional Referral Hospital provides promotive, preventive, curative, and rehabilitative services, including surgical, antenatal and delivery services, pediatrics, internal medicine, and dental care. In addition, the hospital plays a vital role in medical training, research, and supervision of general hospitals and lower-tier health facilities within the Acholi sub-region. GRRH admits an average of 20,934 patients annually, performs 1,997 major operations, and conducts 5,640 deliveries [38]. Every year, the Reproductive Health Department of the hospital provides maternal care to more than 10,000 expectant mothers [39].

## Study population and sampling procedures

The study included pregnant women attending ANC at GRRH who were sampled during the study period. The recruitment of participants was from 6th May 2024–28th May 2024. Women with pregnancy complications, known HBV infection status, or severe allergic reactions to previous vaccine doses were excluded. The sample size was calculated based on the Kish-Leslie formula for cross-sectional studies [40]. At a 95% confidence level, with a z-score of 1.96, a 5% margin of error, a 10% non-response rate, and an estimated prevalence of 50%, a sample size of 430 was determined. The study team used a systematic sampling technique to select the study participants. Every day of data collection, the team developed a list of all pregnant women attending ANC from the clinic's daily attendance record book. Daily, there were an average of 120 women attending the ANC clinic. We used a sampling interval of 10 participants. The first participant was selected by simple random sampling from the list of the first ten participants. Thereafter, every 10th potential participant would be approached and requested to participate in the study. If that potential participant declines to participate, the next person in the queue would be considered. This process was continued every day of data collection until the desired sample size was achieved.

## Data collection tools and procedures

Data were collected using a pre-tested structured questionnaire designed with the KoboCollect mobile software installed on phones and tablets. Data were collected on socio-demographic factors, vaccine-related factors, awareness and knowledge of HBV, perceptions about the vaccine, perceived vulnerability, and health system factors. Data collection was conducted by four trained female research assistants, who had bachelor's degrees in a health-related field, education, or social sciences. The research assistants were recruited and trained for two days on the research objectives and procedures, administration of the study tools, and ethical practices. During training, we involved the research assistants in mock interviews to ensure consistency and uniformity in data collection procedures. After the training, the research assistants participated in the pre-testing of the tools which involved interviews with 20 pregnant women attending ANC at Kawempe Regional Referral Hospital in Kampala. One research assistant entered the data, which was analyzed by a biostatistician, and the investigators refined the tool accordingly. The questionnaire was translated into Acholi to facilitate communication. Acholi is the most spoken language in Gulu city and district. Verification of HBV vaccination for pregnant mothers was done through vaccination cards or doctors' records. During the actual data collection, the research assistants were closely supervised by MLA, who conducted daily debriefing meetings with the research assistants to address challenges encountered during data collection. MLA provided immediate feedback to the research assistants to resolve any emerging challenges and ensure quality data.

## Study variables and measurements

The primary study outcome variable was HBV vaccine uptake which was recorded as a binary variable (Yes = 1, No = 0) based on participants' responses. Vaccination status for pregnant mothers who had received any of the HBV doses, either before or during pregnancy, was verified through vaccination cards or doctors' records. Those with no vaccination were classified as not vaccinated following the criteria used in similar studies [41,42]. The explanatory variables included socio-demographic characteristics (i.e., sex, age, marital status, occupation, tribe, religion, education level), vaccine-related factors (number of doses received, vaccine side effects), awareness and knowledge of HBV, perceptions of the vaccine and perceived vulnerability, and health system factors (screening services, vaccine availability, facility accessibility, availability of health workers, ANC attendance, and health worker advice). Age was collected as a continuous variable and measured in terms of years. Level of education was categorized as primary, secondary, tertiary, and none (no formal education). Sex was measured as either male or female, and religion was categorized into catholic, protestant, Muslim, Pentecostal/born-again, Seventh Day Adventist (SDA), or other. Occupation was categorized into business lady, civil servant, housewife/unemployed, and peasant farmer.

The knowledge variable was assessed by a set of 10 questions including [1] Have you heard about HBV (Yes = 1, No = 0), [2] What is HBV(A liver infection caused by a virus = 1, Others = 0), [3] How is HBV virus transmitted (Through sharing needles or syringes = 1, From mother to child at birth = 1, Through unprotected sexual contact = 1, Others = 0), [4] Which of the following are the symptoms of HBV(Severe headache = 1, Jaundice (yellowing of the skin and eyes)=1, Frequent urination = 1, Rash = 1, Others = 0), [5] Who is at higher risk of developing chronic HBV infection (Infants infected at birth = 1, Healthcare providers and emergency providers = 1, pregnant mothers = 1, sexually active individuals (more than one partner in the last 6 months)=1, men who have sex with fellow men = 1, individuals diagnosed with an STD = 1, illicit drug users = 1 and others = 0), [6] Which of the following is the most effective way to prevent HBV(Vaccination = 1, Others = 0), [7] At what stage of pregnancy is the HBV vaccine recommended (First trimester = 1, Second trimester = 1, Third trimester = 0, at any stage of pregnancy = 0), [8] How many doses of the HBV vaccine are required for complete protection (three doses = 1, Other number of doses = 0), [9] Is the HBV vaccine safe for pregnant women (yes = 1, no = 0), and [10] If a pregnant woman has Hepatitis B, which of the following steps can help prevent transmission to the baby (Delivering the baby by cesarean section only = 1, Vaccinating the baby within 12 hours of birth = 1, Avoiding breastfeeding = 1, Giving the baby a special diet = 0, and others = 0). A maximum total score of 16.0 and a minimum total score of 0.0 were expected to be obtained by the respondents. The median score was obtained and used as a threshold to differentiate between poor and good knowledge of HBV vaccination [43,44]. Respondents who scored below the median were classified as having poor knowledge while those who scored at or above the median were classified as having good knowledge of HBV vaccination.

## Data management and analysis

Data were downloaded from the Kobo Collect server and cleaned using a combination of Microsoft Excel and STATA version 14.0. Descriptive statistics, including frequencies, were used to summarize the data. We used 'modified' Poisson regression to determine the factors associated with the outcome variable. Since the prevalence of HBV vaccine uptake was greater than 10%, we used the prevalence ratio, as using odds ratios would overestimate the strength of the association [45,46]. All variables whose p-values were less than 0.2 were included in the multivariable model using a 'modified' Poisson regression model [47]. Adjusted prevalence ratios at a 95% confidence interval were used to measure the strength of the association and variables whose p-values were less than 0.05 were statistically significant. Variables that were found to be statistically significant (p-value < 0.2) in the bivariate analysis were included in the initial model. The backward stepwise elimination method was used, whereby variables that did not contribute significantly to the model were removed one by one. A multivariable regression model was established, and multicollinearity was assessed. The goodness-of-fit for the final model was assessed using the Pearson chi-square goodness-of-fit test.

## Ethical approval and consent to participate

Ethical approval was obtained from the Research and Ethics Committee of Makerere University School of Public Health (MaKSPH-REC 379) and the Uganda National Council for Science and Technology. Permission to conduct the study was granted by the Hospital Administration. Data collectors obtained written informed consent from each participant after clearly explaining the study's purpose and objectives. Participant confidentiality and privacy were strictly maintained, as no names appeared on the questionnaires, which were completed solely by the participants.

## Results

### Socio-demographic characteristics

A total of 430 participants were enrolled in the study. More than a quarter 28.8% (124/430) were aged 25–29 years, 47.2% (203/430) had attained secondary level education and 36.7% (158/430) were housewives/unemployed. Nearly half, 48.6% (209/430) of the participants were Catholics, 45.8% (197/430) were married, and 70.0% (301/430) were Acholi (Table 1).

**Table 1. Socio-demographic characteristics of participants.**

| Variable | Category | Frequency (N = 430) | Percentage (%) |
|---|---|---|---|
| Age (Mean = 26.0, SD=+/-5.9) | 15-19 years | 48 | 11.2 |
| | 20-24 years | 158 | 36.7 |
| | 25-29 years | 124 | 28.8 |
| | 30 and above years | 100 | 23.3 |
| Level of education | No formal education | 18 | 4.2 |
| | Primary | 162 | 37.7 |
| | Secondary | 203 | 47.2 |
| | Tertiary | 47 | 10.9 |
| Main occupation | Business lady | 146 | 34.0 |
| | Civil servant | 38 | 8.8 |
| | Housewife/ unemployed | 158 | 36.7 |
| | Others | 21 | 4.9 |
| | Peasant farmer | 67 | 15.6 |
| Marital status | Cohabiting/ living together | 180 | 41.9 |
| | Divorced/ Separated | 11 | 2.6 |
| | Married | 197 | 45.8 |
| | Never married/ single | 42 | 9.8 |
| Religion | Anglican | 39 | 9.1 |
| | Catholic | 209 | 48.6 |
| | Muslim | 32 | 7.4 |
| | Pentecost/ Born again | 129 | 30.0 |
| | Others | 6 | 1.4 |
| | SDA | 15 | 3.5 |
| Tribe | Acholi | 301 | 70.0 |
| | Alur | 36 | 8.4 |
| | Others | 93 | 21.6 |

Key: SDA = Seventh Day Adventists

### Knowledge about HBV transmission

More than half, 54.7% (235/430) of the participants were knowledgeable about the HBV virus and its transmission. The majority, 80.0% (344/430) of the participants had heard of HBV, and 46.0% (198/430) mentioned that HBV was transmitted through unprotected sexual contact. More than half, 59.3% (255/430) of the participants mentioned jaundice as one of the symptoms of HBV, over a third (36.3%; 156/430) mentioned that pregnant mothers were at a high risk of developing HBV infection, and 71.2% (306/430) mentioned that vaccination was the most effective way to prevent Hepatitis (Table 2).

### HBV vaccination and infection status

More than half, 53.3% (229/430) of the participants had received the HBV vaccine, and about 39.3% (90/229) had received three doses. More than a third, 39.3% (169/430) of the participants had not been screened for HBV, and a few, 3.0% (13/169) didn't get screened for HBV because the health facilities didn't offer screening services (Table 3).

**Table 2. Knowledge of HBV transmission among pregnant mothers attending ANC at GRRH.**

| Variable | Category | Frequency (N = 430) | Percentage (%) |
|---|---|---|---|
| Heard of HBV | No | 86 | 20.0 |
| | Yes | 344 | 80.0 |
| Causes of Hep (n = 344) | A liver disease caused by bacteria | 124 | 28.8 |
| | A liver infection caused by a virus | 191 | 44.4 |
| | A stomach virus | 16 | 3.7 |
| | A type of cancer | 13 | 3.0 |
| Transmission of HBV* | Through contaminated food | 27 | 6.3 |
| | Through needles | 144 | 33.5 |
| | Mother to child during birth | 151 | 35.1 |
| | Through unprotected sexual contact | 198 | 46.0 |
| Symptoms of HBV* | Severe Headache | 132 | 30.7 |
| | Jaundice (yellowing of the eyes and skin) | 255 | 59.3 |
| | Frequent urination | 56 | 13.0 |
| | Rash | 33 | 7.7 |
| | Others | 54 | 12.6 |
| People at a high risk of developing HBV infection* | Adults over 40 years | 137 | 31.9 |
| | Infants affected at birth | 154 | 35.8 |
| | Healthcare providers and emergency providers | 78 | 18.1 |
| | All pregnant mothers | 156 | 36.3 |
| | Sexually active individuals (more than one partner in the last 6 months) | 121 | 28.1 |
| | Men who have sex with fellow men | 105 | 24.4 |
| | Individuals diagnosed with an STD | 78 | 18.1 |
| | Illicit drug users (injecting, inhaling, snorting, pill-popping) | 63 | 14.7 |
| | Others | 6 | 1.4 |
| Most effective way to prevent HBV (n = 344) * | Antibiotics | 12 | 2.8 |
| | Drinking boiled water | 6 | 1.4 |
| | Regular exercise | 20 | 4.7 |
| | Vaccination | 306 | 71.2 |
| Stage of pregnancy HBV vaccine is recommended (n = 344) | At any stage of pregnancy | 135 | 31.4 |
| | First trimester | 150 | 34.9 |
| | Second trimester | 36 | 8.4 |
| | Third trimester | 23 | 5.3 |
| Number of doses for complete protection (n = 344) | More or less than three | 100 | 23.3 |
| | Three | 244 | 56.7 |
| The HBV vaccine is safe for pregnant women (n = 344) | Don't Know | 95 | 22.1 |
| | No | 84 | 19.5 |
| | Only during the first trimester | 15 | 3.5 |
| | Yes | 150 | 34.9 |
| Transmission of HBV virus to a baby by a pregnant woman | Avoiding breastfeeding | 27 | 6.3 |
| | Delivering the baby by cesarean section only | 18 | 4.2 |
| | Giving the baby a special diet | 14 | 3.3 |
| | Vaccinating the baby within 12 hours of birth | 285 | 66.3 |

**\*Multiple responses allowed**

**Table 3. HBV Vaccination and infection status of participants.**

| Variable | Category | Frequency (N=430) | Percentage (%) |
|---|---|---|---|
| Received HBV vaccine | No | 201 | 46.7 |
| | Yes | 229 | 53.3 |
| Stage of current pregnancy one received the vaccine (n=229) * | First trimester | 69 | 16.0 |
| | I was vaccinated before the pregnancy | 145 | 33.7 |
| | Second trimester | 13 | 3.0 |
| | Third trimester | 2 | 0.5 |
| Number of doses received (n=229) * | I am not sure | 5 | 2.2 |
| | One dose | 49 | 21.4 |
| | Three doses | 90 | 39.3 |
| | Two doses | 85 | 37.1 |
| Reasons for not being vaccinated (n=201) * | I don't have time | 82 | 19.1 |
| | Vaccines are not always available | 38 | 8.8 |
| | Got allergic reactions | 3 | 0.7 |
| | Others | 87 | 20.2 |
| Venue where one received HBV vaccination (n=229) * | At a community health center | 71 | 16.5 |
| | At a different hospital/clinic | 55 | 12.8 |
| | At this hospital/clinic | 88 | 20.5 |
| | Other | 15 | 3.5 |
| Reasons for being vaccinated (n=229) * | Protect the baby from HBV | 70 | 16.3 |
| | My doctor recommended it | 56 | 13.0 |
| | Requirement for receiving ANC | 3 | 0.7 |
| | Concern about my health | 151 | 35.1 |
| | Family/friend recommendation | 28 | 6.5 |
| | Others | 1 | 0.2 |
| Reasons for not being vaccinated (n=201) * | Concerns about the vaccine's safety | 42 | 9.8 |
| | Lack of information about the vaccine | 147 | 34.2 |
| | Fear of needles | 19 | 4.4 |
| | My doctor did not recommend it | 11 | 2.6 |
| | religious beliefs | 2 | 0.5 |
| | Other | 32 | 7.4 |
| Where one heard of HBV (n=229) * | Doctor/nurse at the hospital/clinic | 141 | 32.8 |
| | Television/Radio | 127 | 29.5 |
| | Social media | 19 | 4.4 |
| | Family or friends | 40 | 9.3 |
| | Other | 5 | 1.2 |
| Screened for HBV | No | 169 | 39.3 |
| | Yes | 261 | 60.7 |
| Reasons for not screening HBV (n=169) * | I don't have any signs of the infection | 115 | 26.7 |
| | I don't have time | 32 | 7.4 |
| | Health facilities do have screening services | 13 | 3.0 |
| | Others | 21 | 4.9 |

**\*Multiple responses allowed**

 

## Health system factors

The majority, 79.3% (341/430) of the participants reported that the HBV vaccine was always readily available at health facility when needed. More than a third of the participants 37.9% (163/430) mentioned that it was very easy to access the healthcare facility where the vaccine was available. About 40.2% (173/430) were very satisfied with the information provided by healthcare workers about the HBV vaccine, and 46.0% (198/430) reported that Healthcare workers thoroughly explained the importance of HBV vaccination for them and their babies. The majority, 79.1% (340/430) reported that the health facility offered screening services for HBV (Table 4).

## Factor associated with HBV vaccine uptake among pregnant mothers attending ANC at GRRH

At multivariable analysis, several factors were statistically associated with HBV vaccine uptake among pregnant mothers attending ANC at GRRH. Pregnant mothers aged 30 years and above (APR = 1.10, 95% CI: 1.02–1.18); SDA pregnant mothers (APR = 0.91 95% CI: 0.83–0.99); Those who were knowledgeable about HBV virus, (APR = 1.06 95% CI: 1.01–1.10); The vaccine was available (APR = 1.07, 95% CI: 1.00–1.14); Those who received thorough explanations about the

Table 4. Health system factors regarding HBV services.

| Variable | Category | Frequency (N = 430) | Percentage (%) |
|---|---|---|---|
| The HBV vaccine is always readily available when needed | No | 89 | 20.7 |
| | Yes | 341 | 79.3 |
| Access the healthcare facility where the HBV vaccine is available | Neither easy nor difficult | 62 | 14.4 |
| | Somewhat difficult | 42 | 9.8 |
| | Somewhat easy | 97 | 22.6 |
| | Very difficult | 66 | 15.3 |
| | Very easy | 163 | 37.9 |
| I have been told the HBV vaccine is not available when needed | No | 327 | 76.1 |
| | Yes | 103 | 24.0 |
| Health workers are always around to attend to me | No | 72 | 16.7 |
| | Yes | 358 | 83.3 |
| Health workers offered health education on HBV | No | 147 | 34.2 |
| | Yes | 283 | 65.8 |
| Level of satisfaction with the information provided by healthcare workers about the HBV vaccine | Neither satisfied nor dissatisfied | 38 | 8.8 |
| | Somewhat dissatisfied | 29 | 6.7 |
| | Somewhat satisfied | 87 | 20.2 |
| | Very dissatisfied | 103 | 24.0 |
| | Very satisfied | 173 | 40.2 |
| Healthcare workers explained the importance of HBV vaccination for you and your baby | No | 139 | 32.3 |
| | Yes, but not thoroughly | 93 | 21.6 |
| | Yes, thoroughly | 198 | 46.0 |
| Health facilities offered screening services for HBV | No | 90 | 20.9 |
| | Yes | 340 | 79.1 |
| Ever experienced long waiting times at the healthcare facility for vaccination | Never | 233 | 54.2 |
| | Rarely | 33 | 7.7 |
| | Sometimes | 52 | 12.1 |
| | Yes, always | 112 | 26.0 |

importance of HBV vaccination (APR = 1.17, 95% CI: 1.04–1.32); Pregnant mothers who rarely experienced long waiting times (APR = 1.47, 95% CI: 1.36–1.59), and Pregnant mothers who always experienced long waiting times (APR = 1.48, 95% CI: 1.37–1.59) (Table 5).

## Discussion

We found that the prevalence of HBV vaccine uptake among pregnant women attending ANC at GRRH who had completed the full three doses of HBV vaccine was low, at 20.9%. However, half the women 53.3% had received at least one shot of the vaccine. Our study indicates a lower prevalence of complete vaccination uptake compared to uptakes reported in other studies; for example, vaccine uptake was 33% among pregnant women in Hong Kong [48], and 2.1 per 1000 pregnancies in the USA [49]. The low prevalence of complete vaccination uptake in our study may be attributed to poor access to health-care services. While the higher prevalence of vaccine uptake in Hong Kong and the USA was attributed to the differences in income level, national immunization policies, and routine vaccination programs, and proper health-seeking behav-iors, including antenatal visits of more than 8 times by pregnant mothers. During these visits, the mothers were routinely screened for HBV and encountered healthcare providers who advised them to get vaccinated. Additionally, the vaccination programs for pregnant mothers were free of charge, which could have contributed to a higher vaccination rate.

This study revealed that pregnant mothers aged 30 years and above had higher HBV vaccine uptake compared to those aged 20–24 years. This could be attributed to various factors, such as the perceived risk of HBV transmission among older mothers to their infants, older mothers having stronger social support networks that encourage vaccination, and having more frequent interactions with healthcare providers due to regular prenatal visits, which might provide them opportunities to discuss vaccination concerns. A study by Katamba, Mukunya [50] revealed that women who perceived themselves to be at risk of HBV infection were more likely to utilize HBV screening services compared to those who per-ceived themselves not to be at risk of HBV infection. This, therefore, calls for the promotion of vaccination among preg-nant mothers, regardless of age, through mass sensitization and vaccination promotion programmes.

This study showed that SDA participants were more likely to uptake HBV vaccination compared to other Christians. This finding could be attributed to the SDA's emphasis on health and wellness, which includes a proactive approach to preventive healthcare measures such as vaccinations [51]. In scripture and the writings of Ellen G. White, the SDA generally follows health practices advocated by the church, which promotes a healthy lifestyle and preventive care. This religious group often has better access to health education and resources within their community, fostering an environ-ment that encourages vaccination [52,53]. In contrast, other Christian groups might not have the same level of strong institutional emphasis on preventive healthcare, leading to lower vaccination rates. The findings depict a critical element that religious beliefs can play in influencing health behaviors. However, these findings contradict a qualitative study by Lohiniva, Nurzhynska [54] where Seventh-Day Adventist members refused vaccines as needles used involved blood.

The prevalence of HBV vaccine uptake was higher among those who were knowledgeable about the HBV virus, its transmission, and prevention, compared to those who weren't. This trend may stem from informed pregnant mothers who understand the importance of vaccination, recognize HBV risks, and appreciate the vaccine's protective benefits. This finding was similar to that of a study by Kajungu, Muhoozi [55], which showed women familiar with the importance of maternal vaccines had positive attitudes and expressed willingness to take them. This suggests that more health educa-tion should be given to pregnant mothers to sensitize them about the importance of vaccination. Furthermore, pregnant mothers who were told that the vaccine was available had a higher HBV vaccine uptake compared to those who didn't know. Being aware that the vaccine is available during healthcare visits enhances the chances of vaccination, eliminates the need for separate visits and makes it more convenient for pregnant mothers. This is in agreement with findings of a mixed-method study in Kenya, where the unavailability of the vaccine was the primary reason why participants did not receive the recommended doses [56]. Therefore, ensuring vaccine availability at health facilities and effective communica-tion with pregnant mothers are essential for promoting vaccination coverage.

**Table 5. Factor associated with HBV vaccine uptake among pregnant mothers attending ANC at GRRH.**

| Variable | Overall N (%) | Hepatitis B vaccine uptake | | CPR (95% CI) | APR (95% CI) |
| --- | --- | --- | --- | --- | --- |
| | | No (n = 201) | Yes (n = 229) | | |
| **SOCIO DEMOGRAPHICS** | | | | | |
| **Age category (years)** | | | | | |
| 15-19 | 48 (11.2) | 33 (68.8) | 15 (31.3) | 1 | 1 |
| 20-24 | 158 (36.7) | 71 (44.9) | 87 (55.1) | 1.18 (1.06-1.32) | 1.06 (0.99-1.14) |
| 25-29 | 124 (28.8) | 59 (47.6) | 65 (52.4) | 1.16 (1.03-1.30) | 1.03 (0.96-1.10) |
| 30 and above | 100(23.3) | 38 (38.0) | 62 (62.0) | 1.23 (1.10-1.39) | **1.10 (1.02-1.18)** |
| **Education level** | | | | | |
| No formal education | 18 (4.2) | 13 (72.2) | 5 (27.8) | 1 | 1 |
| Primary | 162 (37.7) | 84 (51.9) | 78 (48.1) | 1.16 (0.97-1.37) | 0.94 (0.83-1.08) |
| Secondary | 203 (47.2) | 87 (42.9) | 116 (57.1) | 1.23 (1.04-1.45) | 0.91 (0.80-1.03) |
| Tertiary | 47 (10.9) | 17 (36.2) | 30 (63.8) | 1.28 (1.07-1.54) | 0.97 (0.84-1.12) |
| **Main occupation** | | | | | |
| Business lady | 146 (34.0) | 67 (45.9) | 79 (54.1) | 1 | 1 |
| Civil servant | 38 (8.8) | 10 (26.3) | 28 (73.7) | 1.13 (1.02-1.24) | 0.97 (0.90-1.04) |
| Housewife/ unemployed | 158 (36.7) | 84 (53.2) | 74 (46.8) | 0.95 (0.88- 1.03) | 1.03 (0.98-1.08) |
| Others | 21 (4.9) | 8 (38.1) | 13 (61.9) | 1.05 (0.91-1.21) | 1.06 (0.95-1.18) |
| Peasant farmer | 67 (15.6) | 32 (47.8) | 35 (52.2) | 0.98 (0.90-1.09) | 1.00 (0.93-1.05) |
| **Marital Status** | | | | | |
| Cohabiting/ Living together | 180 (41.9) | 81 (45.0) | 99 (55.0) | 1 | |
| Divorced/Separated | 11 (2.6) | 6 (54.5) | 5 (45.5) | 0.94 (0.76- 1.16) | |
| Married | 197 (45.8) | 92 (46.7) | 105 (53.3) | 0.99 (0.93-1.06) | |
| Never married/single | 42 (9.8) | 22 (52.4) | 20 (47.6) | 0.95 (0.85- 1.07) | |
| **Religion** | | | | | |
| Anglican | 39 (9.1) | 12 (30.8) | 27 (69.2) | 1 | 1 |
| Catholic | 209 (48.6) | 104 (49.8) | 105 (50.2) | 0.89 (0.81-0.98) | 0.93 (0.86-1.01) |
| Muslim | 32 (7.4) | 15 (46.9) | 17 (53.1) | 0.90 (0.79-1.04) | 0.97 (0.87-1.08) |
| Others | 6 (1.4) | 5 (83.3) | 1 (16.7) | 0.69 (0.53-0.90) | 0.89 (0.69-1.13) |
| Pentecost/ Born again | 129 (30.0) | 55 (42.6) | 74 (57.4) | 0.93 (0.84-1.02) | 0.98 (0.90-1.06) |
| SDA | 15 (3.5) | 10 (66.7) | 5 (33.3) | 0.79 (0.65-0.96) | **0.91 (0.83-0.99)** |
| **Tribe** | | | | | |
| Acholi | 301 (70.0) | 134 (44.5) | 167 (55.5) | 1 | |
| Alur | 36 (8.4) | 20 (55.6) | 16 (44.4) | 0.93 (0.83-1.05) | |
| Others | 93 (21.6) | 47 (50.5) | 46 (49.5) | 0.96 (0.89-1.03) | |
| **KNOWLEDGE OF HEPATITIS B VACCINE** | | | | | |
| Poor | 195 (45.3) | 125 (64.1) | 70 (35.9) | 1 | 1 |
| Good | 235 (54.7) | 76 (32.3) | 159 (67.7) | 1.23 (1.16-1.31) | **1.06 (1.01-1.10)** |
| **VACCINATION HISTORY, SCREENING AND DIAGNOSIS** | | | | | |
| **Screening for Hepatitis B** | | | | | |
| No | 169 (39.3) | 123 (72.8) | 46 (27.2) | 1 | 1 |
| Yes | 261 (60.7) | 78 (29.9) | 183 (70.1) | 1.34 (1.26-1.42) | 1.03 (0.97-1.09) |
| **HEALTH SYSTEM FACTORS** | | | | | |
| **Availability of the vaccine when needed** | | | | | |
| No | 89 (20.7) | 78 (87.6) | 11 (12.4) | 1 | 1 |
| Yes | 341 (79.3) | 123 (36.1) | 218 (63.9) | 1.46 (1.36-1.56) | 1.07 (0.98-1.18) |

*(Continued)*

**Table 5.** (Continued)

| Variable | Overall N (%) | Hepatitis B vaccine uptake | | CPR (95% CI) | APR (95% CI) |
|---|---|---|---|---|---|
| | | No (n=201) | Yes (n=229) | | |
| **Accessibility of the vaccine when needed** | | | | | |
| Neither easy nor difficult | 62 (14.4) | 41 (66.1) | 21 (33.9) | 1 | 1 |
| Somewhat difficult | 42 (9.8) | 27 (64.3) | 15 (35.7) | 1.01 (0.88-1.16) | 1.01 (0.90-1.13) |
| Somewhat easy | 97 (22.6) | 34 (35.1) | 63 (64.9) | 1.23 (1.11-1.37) | 1.01 (0.93-1.10) |
| Very difficult | 66 (15.3) | 55 (83.3) | 11 (16.7) | 0.87 (0.78- 0.98) | 0.98 (0.89-1.09) |
| Very easy | 163 (37.9) | 44 (27.0) | 119 (73.0) | 1.29 (1.17-1.42) | 0.97 (0.90-1.04) |
| **Ever been told that the Hepatitis B vaccine was not available when one went to get vaccinated** | | | | | |
| Never asked about vaccine availability | 106 (24.7) | 62 (58.5) | 44 (41.5) | 1 | 1 |
| No | 221 (51.4) | 130 (58.8) | 91 (41.2) | 1.00 (0.92- 1.08) | 0.99 (0.93-1.05) |
| Yes | 103 (24.0) | 9 (8.7) | 94 (91.3) | 1.35 (1.26- 1.45) | **1.07 (1.00-1.14)** |
| **Health workers are always around to attend to you** | | | | | |
| No | 72 (16.7) | 64 (88.9) | 8 (11.1) | 1 | 1 |
| Yes | 358 (83.3) | 137 (38.3) | 221 (61.7) | 1.46 (1.35- 1.56) | 1.09 (0.97-1.23) |
| **Health workers offered health education on Hepatitis B** | | | | | |
| No | 147 (34.2) | 122 (83.0) | 25 (17.0) | 1 | 1 |
| Yes | 283 (65.8) | 79 (27.9) | 204 (72.1) | 1.47 (1.38-1.56) | 0.96 (0.85-1.09) |
| **Level of satisfaction with the information provided by healthcare workers about the Hepatitis B vaccine** | | | | | |
| Neither satisfied nor dissatisfied | 38 (8.8) | 29 (76.3) | 9 (23.7) | 1 | 1 |
| Somewhat dissatisfied | 29 (6.7) | 19 (65.5) | 10 (34.5) | 1.09 (0.92- 1.29) | 1.07 (0.92-1.24) |
| Somewhat satisfied | 87 (20.2) | 34 (39.1) | 53 (60.9) | 1.30 (1.15- 1.48) | 1.03 (0.92-1.16) |
| Very dissatisfied | 103 (24.0) | 87 (84.5) | 16 (15.5) | 0.93 (0.82- 1.06) | 1.07 (0.94-1.22) |
| Very satisfied | 173 (40.2) | 32 (18.5) | 141 (81.5) | 1.47 (1.31- 1.64) | 1.10 (0.98-1.24) |
| **Healthcare workers explained the importance of Hepatitis B vaccination for you and your baby** | | | | | |
| No | 139 (32.3) | 118 (84.9) | 21 (15.1) | 1 | 1 |
| Yes, but not thoroughly | 93 (21.6) | 36 (38.7) | 57 (61.3) | 1.40 (1.29-1.51) | **1.17 (1.04-1.32)** |
| Yes, thoroughly | 198 (46.0) | 47 (23.7) | 151 (76.3) | 1.53 (1.44-1.63) | **1.17 (1.04-1.31)** |
| **The health facility offered screening services for Hepatitis B** | | | | | |
| No | 90 (20.9) | 76 (84.4) | 14 (15.6) | 1 | 1 |
| Yes | 340 (79.1) | 125 (36.8) | 215 (63.2) | 1.41 (1.31-1.52) | 0.93 (0.83-1.03) |
| **Experienced long waiting times at the healthcare facility for vaccination** | | | | | |
| Never | 233 (54.2) | 190 (81.5) | 43 (18.5) | 1 | 1 |
| Rarely | 33 (7.7) | 4 (12.1) | 29 (87.9) | 1.59 (1.47-1.71) | **1.47 (1.36-1.59)** |
| Sometimes | 52 (12.1) | 2 (3.8) | 50 (96.2) | 1.66 (1.58-1.74) | **1.50 (1.40-1.62)** |
| Yes, always | 112 (26.0) | 5 (4.5) | 107 (95.5) | 1.65 (1.57-1.72) | **1.48 (1.37-1.59)** |

CPR = Crude prevalent ratios; APR = Adjusted Prevalence ratios.

Adjustment done for age, religion, education level and knowledge about Hepatitis B.

The study also found that pregnant mothers who received clear explanations about the importance of HBV vaccination had higher vaccine uptake. In general, pregnant women are likely to undertake vaccinations when healthcare professionals give them adequate explanations regarding the nature and benefits of the vaccination. Clear explanations address doubts, myths, and misconceptions about vaccination, thus motivating mothers to make well-informed health decisions. Therefore, healthcare providers should allocate adequate time for health education and flexible scheduling options to accommodate pregnant individuals' needs and use educational videos and media campaigns to deliver clear and

consistent messages. These findings are in agreement with a systematic review on vaccine uptake among pregnant mothers which revealed that clear communication on the benefits and safety of immunization, and positive social influences from family and friends [57] are key factors that influence mothers uptake of vaccinations. Although longer waiting times are typically viewed as a barrier to service utilization, long waiting time at antenatal care clinics has been associated with dissatisfaction with the services. Satisfaction was associated with waiting time of less than 45 minutes (62), our findings suggest that pregnant mothers who rarely, sometimes, or always experienced long waiting times had a higher prevalence of HBV vaccine uptake compared to those who never experienced long waiting times. These mothers who experience extended waits may represent a highly motivated subgroup with strong confidence in immunization services (63). Contrary to a study done by Ecarnot, Crepaldi (61) which revealed that patients reported shorter waiting times was a motivator for vaccination. This underscores the importance of addressing both supply-side efficiency and demand-side determinants when designing immunization programs like appointment scheduling, efficient 26 triage, and clear communication about waiting time should be implemented to increase vaccine uptake.

### Study limitation and strengths

Our study relied on self-reported vaccination status among participants without confirmation through anti-HBV surface antibody (HBsAb) testing, which might have introduced recall bias, however, we verified with hospital records and vaccination cards. Furthermore, the study was prone to social desirability bias. Social desirability bias may have occurred when participants responded, on what they believed were more socially acceptable or favorable, rather than their true practices. However, we addressed this by ensuring participants identities remained confidential when discussing sensitive topics. Additionally, whenever feasible, we relied on objective pre-existing records (such as medical records) rather than relying solely on participant recollections. The study's focus on pregnant women attending ANC at GRRH limits its generalizability to the wider region or country. Nonetheless, the findings provide valuable insights specific to this population and region and can serve as a basis for future, more extensive studies.

### Conclusion

More than half of pregnant mothers had received at least a dose of the HBV vaccination. Key factors associated with uptake include age, religion, knowledge about HBV, healthcare worker communication, availability of the vaccine, and waiting times at the healthcare facilities. There is an urgent need to improve HBV vaccine uptake among pregnant women, and prioritize raising awareness, strengthening communication among healthcare workers, ensuring consistent vaccine availability, and minimizing wait times at health facilities.

### Supporting information

**S1 File. S1 dataset.**
(XLSX)

### Acknowledgments

We thank the GRRH administration offices for granting administrative clearance to conduct the study. Our gratitude also goes to the research assistants who diligently conducted the study and the pregnant mothers within the jurisdiction of GRRH for sparing their time to respond to the survey questions.

### Author contributions

**Conceptualization:** Linda Mercy Akello, Amos Deogratius Mwaka, Christopher Garimoi Orach.

**Data curation:** Linda Mercy Akello, Jimmy Osuret, Jovan Galiwango.

**Formal analysis:** Linda Mercy Akello, Jovan Galiwango.

**Investigation:** Linda Mercy Akello.

**Methodology:** Linda Mercy Akello.

**Resources:** Linda Mercy Akello.

**Supervision:** Jimmy Osuret, Christopher Garimoi Orach.

**Validation:** Linda Mercy Akello, Amos Deogratius Mwaka, Christopher Garimoi Orach.

**Visualization:** Linda Mercy Akello, Jimmy Osuret, Jovan Galiwango, Amos Deogratius Mwaka, Christopher Garimoi Orach.

**Writing – original draft:** Linda Mercy Akello.

**Writing – review & editing:** Linda Mercy Akello.

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
