## [Decision Letter · Decision Letter 0]

16 Mar 2026

PONE-D-26-01864Hepatitis B Vaccine Uptake and Associated Factors Among Pregnant Women Attending Antenatal Care at Gulu Regional Referral Hospital, Northern UgandaPLOS One

Dear Dr. Akello,

Thank you for submitting your manuscript to PLOS ONE. After careful consideration, we feel that it has merit but does not fully meet PLOS ONE’s publication criteria as it currently stands. Therefore, we invite you to submit a revised version of the manuscript that addresses the points raised during the review process.

We look forward to receiving your revised manuscript.

Kind regards,

Madhavi Yennapu, PhD

Academic Editor

PLOS One

Journal Requirements:

4. Please amend the manuscript submission data (via Edit Submission) to include authors Mercy Linda Akello and Christopher Garimoi Orach.

5. Please amend your authorship list in your manuscript file to include authors Linda Mercy Akello and Christopher Orach Garimoi.

Additional Editor Comments:

Dear Mercy Linda Akello,

Your article (PONE-D-26-01864) on 'Hepatitis B Vaccine Uptake and Associated Factors Among Pregnant Women Attending Antenatal Care at Gulu Regional Referral Hospital, Northern Uganda' is interesting and of public health importance. However, both the reviewers suggested a revision of the manuscript, one major and another minor review. Kindly address the revisions and resubmit your manuscript.

Best,

Academic editor

Madhavi

Reviewers' comments:

Reviewer's Responses to Questions

**Comments to the Author**

1. Is the manuscript technically sound, and do the data support the conclusions?

Reviewer #1: Yes

Reviewer #2: Yes

2. Has the statistical analysis been performed appropriately and rigorously? 

Reviewer #1: Yes

Reviewer #2: Yes

3. Have the authors made all data underlying the findings in their manuscript fully available?

Reviewer #1: Yes

Reviewer #2: Yes

4. Is the manuscript presented in an intelligible fashion and written in standard English?

Reviewer #1: Yes

Reviewer #2: Yes

5. Review Comments to the Author

Reviewer #1: 1. Throughout manuscript avoid use of "Hep B" and abbreviate "Hepatitis B virus " as "HBV"

2. Update statistics to 2026 or 2025, the numbers have changed.

3. Include REC/UNCST approval numbers on the ethics sections.

4. What was the mean age of the participants?

5. There is an additional study on HBV in Uganda not cited "Gwokyalya AM, Nakityo I, Twinamasiko N, Kihumuro RB, Isiko I, Kasoma RM, Bongomin F. Hepatitis B Vaccination History among medical intern doctors and nurses in three national referral hospitals in Uganda. Afr Health Sci. 2025 Dec;25(4):68-74. doi: 10.4314/ahs.v25i4.8. PMID: 41669128; PMCID: PMC12883965."

6. Ensure literature review to inform the introduction and discussion sections are uptodate .

Reviewer #2: The manuscript entitled “Hepatitis B Vaccine Uptake and Associated Factors Among Pregnant Women Attending Antenatal Care at Gulu Regional Referral Hospital, Northern Uganda” may be considered for publication after the following revisions.

The study addresses an important public health question and applies appropriate methods; however, clearer reporting, minor analytical refinements, and strengthened interpretation are required.

Major Comments

Definition of Vaccine Uptake

The manuscript defines uptake as “receipt of at least one dose,” yet the abstract and conclusion emphasize “complete vaccination” (three doses). Please ensure consistency throughout the manuscript and clearly distinguish between “any-dose uptake” and “complete series coverage.”

Sampling Strategy

The Methods section describes systematic sampling from ANC attendance lists with replacement of refusals, although the study is described as “randomly selected.” Please clarify the exact sampling approach and revise terminology for accuracy.

Study Population and Generalizability

It should be explicitly stated that the sample represents pregnant women attending ANC at Gulu Regional Referral Hospital (GRRH), not all pregnant women in Northern Uganda. This limitation should be clearly acknowledged in the Discussion.

Knowledge Assessment and Scoring

Knowledge was assessed using 10 items (including multi-response questions), generating a 0–16 score, which was dichotomized at the median into “good” and “poor.” Please clarify:

How multi-response items were scored (e.g., one point per correct option regardless of incorrect selections, or whether incorrect responses were penalized).

The rationale for dichotomizing at the median.

Consider conducting a sensitivity analysis by modeling knowledge as a continuous variable or categorizing into tertiles. At minimum, acknowledge the potential information loss from dichotomization as a limitation.

Regression Model Clarity and Variable Selection

The use of modified Poisson regression with robust variance to estimate prevalence ratios is appropriate. However:

The Methods section inconsistently states that variables with p<0.2 (and elsewhere p<0.05) were included in the multivariable model. Please standardize this criterion.

Table 5 indicates adjustment only for age, religion, education, and knowledge. Clarify whether other covariates were considered and the final model-building strategy.

Results Presentation

Please clearly distinguish:

Any-dose uptake: 53.3%

Complete series coverage: 20.9% among vaccinated women (90/229 ≈ 21%), which corresponds to approximately 21% (90/430) of the total sample.

Present these proportions consistently to avoid confusion.

Contextual Comparison

The comparison with uptake rates in Hong Kong and the United States is appropriate; however, please contextualize these findings by acknowledging differences in income level, national immunization policies, and routine birth-dose/adult vaccination programs to avoid over-interpretation.

Minor Comments

Overall, the English language is generally clear; however, minor grammatical and typographical errors should be corrected through careful proofreading.

6. PLOS authors have the option to publish the peer review history of their article (what does this mean?). If published, this will include your full peer review and any attached files.

Reviewer #1: No

Reviewer #2: **Yes:**Amit Kumar Mittal

---

## [Author Response · Author response to Decision Letter 1]

14 Apr 2026

The Comments have been addressed and the documents attached.

---

## [Decision Letter · Decision Letter 1]

6 May 2026

PONE-D-26-01864R1Hepatitis B Vaccine Uptake and Associated Factors Among Pregnant Women Attending Antenatal Care at Gulu Regional Referral Hospital, Northern UgandaPLOS One

Dear Dr. Akello,

Thank you for submitting your manuscript to PLOS ONE. After careful consideration, we feel that it has merit but does not fully meet PLOS ONE’s publication criteria as it currently stands. Therefore, we invite you to submit a revised version of the manuscript that addresses the points raised during the review process.

Thanks for submitting your revised draft. Both the reviewers have gone through your revised draft, agree on the merits of your draft except on the statistical analysis. Though, one of the reviewers (reviewer2) states ‘I don’t know’, with respect to its statistical analysis, I would like to go by the judgement of Reviewer1, that the statistical analysis has been rigorous and appropriate, based on his credentials in epidemiology.

However, there are still some minor revisions that need to be addressed and taken care of as suggested by both the reviewers. Both the reviewers suggest ‘minor review’ and Kindly do the needful and resubmit again.

We look forward to receiving your revised manuscript.

Kind regards,

Madhavi Yennapu, PhD

Academic Editor

PLOS One

Journal Requirements:

Reviewers' comments:

Reviewer's Responses to Questions

**Comments to the Author**

1. If the authors have adequately addressed your comments raised in a previous round of review and you feel that this manuscript is now acceptable for publication, you may indicate that here to bypass the “Comments to the Author” section, enter your conflict of interest statement in the “Confidential to Editor” section, and submit your "Accept" recommendation.

Reviewer #1: All comments have been addressed

Reviewer #2: All comments have been addressed

2. Is the manuscript technically sound, and do the data support the conclusions?

Reviewer #1: Yes

Reviewer #2: Yes

3. Has the statistical analysis been performed appropriately and rigorously? 

Reviewer #1: Yes

Reviewer #2: I Don't Know

4. Have the authors made all data underlying the findings in their manuscript fully available?

Reviewer #1: Yes

Reviewer #2: Yes

5. Is the manuscript presented in an intelligible fashion and written in standard English?

Reviewer #1: Yes

Reviewer #2: Yes

6. Review Comments to the Author

Reviewer #1: Thank you for addressing all the comments

1. in the abstract HBV is abbreviated as "HPV" kindly correct

2. The reference list has so many web pages of WHO/MoH which could be minimised

3. Use PLoS One referencing style (X), Vancouver

4. In the discussion , ANC didnt need to be spelled out in full, just since already abbreviated earlier on

Reviewer #2: (No Response)

7. PLOS authors have the option to publish the peer review history of their article (what does this mean?). If published, this will include your full peer review and any attached files.

Reviewer #1: **Yes:**Felix Bongomin

Reviewer #2: **Yes:**Amit Kumar MITTAL

---

## [Author Response · Author response to Decision Letter 2]

11 May 2026

All the comments have been addressed and attached in the document titled "response to reviewers"

---

## [Editor Report · Decision Letter 2]

14 May 2026

Hepatitis B Vaccine Uptake and Associated Factors Among Pregnant Women Attending Antenatal Care at Gulu Regional Referral Hospital, Northern Uganda

PONE-D-26-01864R2

Dear Dr. Akello,

We’re pleased to inform you that your manuscript has been judged scientifically suitable for publication and will be formally accepted for publication once it meets all outstanding technical requirements.

Kind regards,

Madhavi Yennapu, PhD

Academic Editor

PLOS One
---

## [Editor Report · Acceptance letter]

PONE-D-26-01864R2

PLOS One

Dear Dr. Akello,

I'm pleased to inform you that your manuscript has been deemed suitable for publication in PLOS One. Congratulations! Your manuscript is now being handed over to our production team.

Kind regards,

on behalf of

Dr. Madhavi Yennapu

Academic Editor

PLOS One